# Gaussia Luciferase as a Reporter for Quorum Sensing in *Staphylococcus aureus*

**DOI:** 10.3390/s20154305

**Published:** 2020-08-01

**Authors:** Isobel Blower, Carmen Tong, Xiaohui Sun, Ewan Murray, Jeni Luckett, Weng Chan, Paul Williams, Philip Hill

**Affiliations:** 1School of Life Sciences, Biodiscovery Institute, University of Nottingham, University Park, Nottingham NG7 2RD, UK; Isobel.Blower@nottingham.ac.uk (I.B.); C.Tong@sygnaturediscovery.com (C.T.); Ewan.Murray@nottingham.ac.uk (E.M.); jeni.luckett@nottingham.ac.uk (J.L.); paul.williams@nottingham.ac.uk (P.W.); 2School of Biosciences, Sutton Bonington Campus, University of Nottingham, Loughborough LE12 5RD, UK; sunxiaohui@hqu.edu.cn; 3College of Chemical Engineering, Huaqiao University, 668 Jimei Blvd, Xiamen 361021, China; 4School of Pharmacy, Biodiscovery Institute, University of Nottingham, University Park, Nottingham NG7 2RD, UK; weng.chan@nottingham.ac.uk

**Keywords:** *Gaussia* luciferase, quorum sensing, *Staphylococcus aureus*, *agr*

## Abstract

*Gaussia* luciferase (GLuc) is a secreted protein with significant potential for use as a reporter of gene expression in bacterial pathogenicity studies. To date there are relatively few examples of its use in bacteriology. In this study we show that GLuc can be functionally expressed in the human pathogen *Staphylococcus aureus* and furthermore show that it can be used as a biosensor for the *agr* quorum sensing (QS) system which employs autoinducing peptides to control virulence. GLuc was linked to the P3 promoter of the *S. aureus*
*agr* operon. Biosensor strains were validated by evaluation of chemical agent-mediated activation and inhibition of *agr*. Use of GLuc enabled quantitative assessment of *agr* activity. This demonstrates the utility of *Gaussia* luciferase for in vitro monitoring of *agr* activation and inhibition.

## 1. Introduction

*Staphylococcus aureus* has been identified by the World Health Organization as a high priority pathogen [1]. It is one of the most common causes of bacterial infections in both hospitals and in the community [2]. Infections caused by *S. aureus* are difficult to treat due to the ability of this bacterium to acquire resistance to multiple antibiotics rapidly and to grow as a biofilm which limits the efficacy of diverse antibiotics and biocides [3,4]. There is, therefore, an urgent need to develop antimicrobials with novel targets which are effective against resistant strains. As well as traditional drugs, alternative therapies are being investigated which are less likely to drive the development of resistance [5]. These include compounds which will target functions essential for infection such as colonization and virulence factor production without affecting bacterial growth [6].

In *S. aureus* the accessory gene regulator (*agr*) quorum sensing system is a key regulator of virulence factor expression [2,7]. Quorum sensing (QS) is a means of cell to cell communication in bacterial populations. It coordinates cooperative behavior by enabling bacteria at high population densities to respond to altered environments [8]. Bacteria communicate using secreted signal molecules called autoinducers. When the extracellular concentration of the autoinducer reaches a threshold level, a signal transduction pathway is activated, leading to targeted gene activation or repression [7].

For the *S. aureus agr* system, as outlined in Figure 1, the quorum sensing signal molecule is an autoinducing peptide (AIP) [9]. AIPs are produced by post-translational N- and C-terminal processing of the AgrD pre-propeptide by AgrB and exported from the cell [7,10]. Once an AIP reaches a threshold concentration outside the cell it binds to and activates AgrC in the cell membrane. This autophosphorylates in response to AIP binding and in turn phosphorylates AgrA, the response regulator. Activated AgrA binds to the P2 and P3 promoters, upregulating *agrBCDA* and RNAIII expression [10,11,12]. RNAIII is the effector of the *agr* system, it encodes δ-haemolysin and also acts as a regulatory RNA that controls expression of multiple virulence genes post-transcriptionally [7].

There is increasing evidence to suggest that blocking *S. aureus* virulence gene expression attenuates infection in certain experimental animal infection models [14,15]. This has been achieved through mutation of *agr* or inhibition of *agr* via various natural and synthetic compounds. Inhibitors of *agr* target different components within this quorum sensing system including those which target AgrB to block AIP generation, competitive AIP inhibitors (also known as AgrC antagonists) which prevent AgrC activation and inhibition of P2 and P3 activation by AgrA (Figure 1) [13]. Attenuating virulence early in infection could allow the host immune system to respond and clear the pathogen, sparing exposure to antibiotics [8]. Virulence inhibitors could also be used as adjuvants with existing antibiotics, or with therapies which boost the host immune response [16,17].

Currently it is unclear when or where staphylococcal *agr* is expressed within the infection site and whether *agr* inhibitors could be used against established infections or as prophylactics. In order to investigate temporal *agr* expression and *agr* inhibition it is important to develop methods for real time monitoring of *agr* activity which can be used in vivo [18]. This can be performed using bioluminescent reporters linked to promoters of the *agr* system [19]. In previous studies, the *Photorhabdus luminescens lux* operon linked to the *agr*P3 promoter has been used to measure temporal *agr* activation in infection models and assess *agr* inhibitors in vivo [14,15]. *agr*P3 *lux* based systems have also been used for high-throughput screening of *agr* inhibitors [12] though it has been reported that some compounds such as *Pseudomonas aeruginosa* quorum sensing molecule 3-oxo-C_12_-HSL directly inhibit Lux-based bioluminescence, potentially leading to false positive hits [20]. Lux-based systems also require oxygen and so are potentially compromised in environmental niches where oxygen levels are low. In this study, a novel bioluminescent reporter linked to the P3 promoter of the *agr* system was validated for use in *S. aureus*.

Luciferases are often used as biological reporters due to their high sensitivity, broad dynamic range and operational simplicity. Luciferases have been used to monitor multiple cellular processes, such as gene expression and protein interactions as well as tracking of bacterial and viral infection [21,22]. Traditionally, bacterial luciferases have been used as reporter genes in staphylococci, however this system requires substrate and energy supply from the cells, at the point of live-cell assay. Novel luciferases such as the *Gaussia* luciferase, which is small, secreted and ATP-independent, broaden the potential applications and provide the ability to detect bioluminescence extracellularly and without the need for extra energy from live cells at the point of the assay [23].

*Gaussia* luciferase (GLuc) is a naturally secreted enzyme from the marine copepod *Gaussia princeps* [24]. GLuc is a 185 amino acid, 19.9 kDa luciferase which catalyses oxidative decarboxylation of the substrate coelenterazine (CTZ) producing coelenteramide (CTM), emitting blue light at 480 nm [25]. GLuc is nontoxic and is naturally secreted from the cell therefore is useful for monitoring biological processes without the requirement for cellular energy to drive the luminescent reaction [26]. GLuc is a useful reporter due to its small size and high light intensity, and hence delivering assays with high sensitivity. GLuc is also stable at a wide range of pHs and is resistant to heat shock and oxidative stress [27]. GLuc has been shown to be an effective biological reporter in bacteria, such as *Salmonella enterica* and *Mycobacterium smegmatis* [26,28]. It is a highly sensitive reporter for nondestructive quantification of in vivo biological processes ex vivo, as bioluminescence can be detected from the blood or urine of in vivo models using a luminometer [29,30].

The aim of this work was to demonstrate the use of *Gaussia* luciferase as a reporter in *S. aureus* and for monitoring activation of the *agr* quorum sensing system (see Figure 1). Reporter strains were constructed with GLuc chromosomally encoded under the control of the *agr*P3 promoter in *S. aureus* USA300 strain JE2 and an isogenic Δ*agr* mutant. These were validated by evaluation of activation and inhibition of the *agr* system using the cognate AIP as agonist and synthetic (Ala^5^)AIP-1 as an antagonist and by testing both nonpeptidic AgrC allosteric and AgrB inhibitors. The data obtained show that GLuc can be used successfully for in vitro monitoring of *agr* activation and inhibition.

## 2. Materials and Methods

Staphylococcal strains were cultured in tryptone soy broth (TSB) (Oxoid) and *E. coli* in lysogeny broth (LB) (Oxoid) with antibiotics added as required. All strains were grown at 37 °C and overnight cultures were grown with aeration by shaking at 200 rpm. AIPs, *N*-(3-oxododecanoyl)-L-homoserine lactone (3-oxo-C_12_-HSL, (S)-3-dodecanoyl-5-(2-hydroxyethyl) tetramic acid (C_12_-TMA) and 3-dodecanoyltetronic acid (C_12_-TOA) were synthesized as described before. Ambuic acid was purchased from Cayman Chemicals.

### 2.1. Construction of agrP3 Reporter S. aureus Strains

A transcriptional reporter fusion was constructed with *Gaussia* encoded chromosomally under the control of the *agr*P3 promoter. All strains and primers used are listed in Table 1 and Table 2, respectively. To ensure GLuc would be secreted by *S. aureus,* the protein A (SpA) leader was engineered upstream of a synthetic GLuc gene (kind gift of S. Wiles, University of Auckland, Auckland, New Zealand). This replaced the sequence of the original native eukaryotic leader to direct the translated protein to the *S. aureus* Sec secretory pathway. This fragment was assembled into pUNKdestR3R4 flanked by the Pxyl/tet promoter and the rrnB1B2 transcriptional terminator to create plasmid pUNK1 pXyl/tet:SpA leader:GLuc using Gateway cloning [31] and provided for constitutive expression of Gluc in *S. aureus*. The GLuc gene was then amplified by PCR from this vector using primer pair CT16 and CT17 using high fidelity Phusion DNA polymerase (Thermofisher). The *agr*P3 promoter was also amplified by PCR from *S. aureus* 8325-4 genomic DNA using primer pair CT14 and CT15 and assembled upstream of GLuc into pUNK1 using Hifi DNA assembly (NEB). This was used to transform electrocompetent *E. coli* DC10B. Plasmids were extracted using a miniprep kit (Qiagen) before confirmation by sequencing (Source Bioscience).

Chromosomal integration of *agr*P3-GLuc into *S. aureus* was performed as outlined by Lei et al. [33] using integration vector pLL102. Plasmids pUNK1 *agr*P3:SpA leader:GLuc and pLL102 were digested with *Sal*I and fragments ligated using T4 DNA ligase (NEB) and used to transform electrocompetent *E. coli* DC10B to create plasmid pLL102 *agr*P3:SpA leader:GLuc. This plasmid was used to transform electrocompetent *S. aureus* RN4220 attB2. Chromosomal integration was checked by colony PCR using primer pairs OU9R10 and SCV4, and SCV8 and OU9R7 and the correct strain named *S. aureus* RN4220 attB2 *agr*P3 SpA leader GLuc. This strain was used as a donor for phage transduction into *S. aureus* USA300 strain JE2 using phage phi11 (Φ11). Further phage transduction was performed with a Δ*agr* Φ11 phage lysate. This resulted in the construction of reporter strains USA300 *agr*P3-GLuc and USA300 Δ*agr* P3 GLuc.

### 2.2. Measuring agrP3 Promoter Activity Using GAUSSIA Luciferase Bioluminescence

When testing bioluminescence output from GLuc over time, a single colony of the test strain was used to inoculate 50 mL of TSB, which was incubated at 37 °C, 200 rpm. An OD_600_ reading of a 1 mL sample was taken every hour for 12 h. The supernatant fraction was separated from the cells by centrifugation twice at 13,000× *g* for 2 min. Samples were stored at −20 °C prior to analysis. Freeze thawing has been shown to have no effect on the stability of *Gaussia* luciferase [38]. When testing the effect of AIP-1 and ambuic acid, a single colony was grown in 5 mL of TSB broth for 16 h with test compounds added at the desired concentrations at the start of incubation. When testing the effect of Ala5(AIP-1) and 3-oxo-C_12_-HSL analogues these compounds were added to broth with a single colony as before and incubated for 6 h. Cultures were incubated at 37 °C at 200 rpm and samples taken as described previously.

Samples were thawed at room temperature and 100 µL loaded into a black, flat clear-bottomed 96 well plate (Greiner). Bioluminescence was recorded as Relative Light Units (RLU) using a 96 well-plate reader and injector (Infinite F200 Tecan): 50 µL of 20 µM coelenterazine (CTZ) substrate was injected at 200 µL/s and bioluminescence read with an integration time of 100 ms after a wait time of 1 s. A portion of 20 µM CTZ was used as this was shown to saturate the enzyme. CTZ was injected into each well and bioluminescence read in turn, 3 technical repeats were recorded for each sample. Statistical analysis was performed using an unpaired *t*-test with Welch’s correction.

## 3. Results

### 3.1. Induction of agr with Respect to Growth Phase

In *S. aureus, agr-*dependent quorum sensing is activated at high cell population densities when the autoinducing AIP reaches a threshold level. This has been reported to occur in vitro in the late exponential/early stationary phase of growth [11]. When the AIP reaches the threshold concentration the *agr*P3 promoter is activated, therefore GLuc expression will be induced via *agr*P3. In order to determine whether the production of GLuc bioluminescence occurs as a function of the growth phase consistent with *agr* induction, bioluminescence and OD_600_ were quantified over 12 h using *S. aureus* USA300 *agr*P3-GLuc.

A single colony was inoculated into 50 mL TSB broth with 0.01% v/v DMSO added as a solvent control. Growth did not appear to be attenuated. As shown in Figure 2a, light output was observed after 6–7 h of growth. From 5–7 h, the level of bioluminescence increased substantially, from ~4.83 × 10^5^ to 8.74 × 10^6^ RLU/OD_600_ (Relative Light Units/Optical Density at 600 nm). The increase in bioluminescence reached a plateau after 10 h at ~2.7 × 10^7^ RLU/OD_600_ which is consistent with *agr* expression.

### 3.2. Activation of USA300 Δagr P3-GLuc pSKermP2 agrAC with Synthetic AIP-1

To confirm the AIP-dependent expression of the *agr*P3-Gluc construct we set out to demonstrate that the reporter can be activated with synthetic, exogenous AIP-1. The *agrA* and *agrC* genes which constitute the *agr* two-component signal transduction system, were introduced into USA300 Δ*agr* P3-GLuc on plasmid pSKermP2 *agr*AC [36]. Genetic complementation with *agrA* and *agrC* would enable the strain to respond to exogenous AIP-1 by upregulation of the *agr*P3-GLuc promoter due to signal transduction from AgrC and AgrA [36]. Due to the lack of AgrB and AgrD the strain could not itself synthesize AIP-1.

The response of this strain to synthetic AIP-1 was demonstrated by the addition of 100 nM AIP-1 to an overnight culture. As a control, 0.01% v/v DMSO, the solvent used for AIP-1 was tested, as well as USA300 Δ*agr* P3-GLuc with addition of 100 nM AIP-1.

As shown in Figure 3 after addition of AIP-1 to USA300 Δ*agr* P3-GLuc the level of bioluminescence was ~6.78 × 10^6^ RLU/OD_600_. This demonstrates that AIP-1 does not activate quorum sensing in the Δ*agr* strain. When complemented with *agrA* and *agrC*, the 0.01% v/v DMSO control produced bioluminescence of ~1.4 × 10^7^ RLU/OD_600_, indicating the levels of background luminescence when the AgrA transcription activator was present. On exogenous provision of 100 nM AIP-1 this increased to ~4.19 × 10^7^ RLU/OD_600_. This was a significant increase showing the strain responded productively to exogenous AIP-1.

### 3.3. Inhibition of S. aureus USA300 agrP3 by an AgrC Antagonist (Ala^5^)AIP-1

To demonstrate that GLuc expression is under the control of the *agr*P3 promoter, *S. aureus* USA300 *agr*P3-GLuc was treated with (Ala^5^)AIP-1, a competitive inhibitor of the *agr* system [39]. Bioluminescence was recorded after incubation for 6 h. As shown in Figure 4, after 6 h there was a significant decrease in bioluminescence observed when treated with 100 nM (Ala^5^)AIP-1 (~6.4 × 10^5^ RLU) compared with the 1% v/v DMSO control (1.13 × 10^6^ RLU). This demonstrates that bioluminescence from this strain can be inhibited with addition of a competitive antagonist of the *agr* system.

### 3.4. Evaluation of an AgrB Inhibitor with USA300 agrP3-GLuc

Inhibitors of other targets in the *agr* system have been identified, such as ambuic acid which blocks AIP biosynthesis by targeting AgrB [15]. This was tested against USA300 *agr*P3-GLuc. Addition of 40 µM ambuic acid significantly reduced bioluminescence compared with the control to 2.52 × 10^7^ RLU/OD_600_ (Figure 5). This was almost 4-fold lower than the control although bioluminescence was not reduced to the level of the *agr* mutant.

### 3.5. 3-Oxo-C_12_-HSL Has Differential Effects on GLuc and Lux Bioluminescence

The molecule 3-Oxo-C_12_-HSL is a *Pseudomonas aeruginosa* quorum sensing signal molecule which has inhibitory effects on the *S. aureus agr* quorum sensing system [40]. Murray et al. screened analogues of 3-oxo-C_12_-HSL and identified C_12_-TMA and C_12_-tetronic acid (C_12_-TOA) as the most potent inhibitors. They used a *blaZ* based *agrP3* reporter as 3-oxo-C12-HSL affected *lux*-based bioluminescence making conventional *lux-*based *agr*P3 reporter systems unsuitable [40].

To determine whether a GLuc-based reporter could be used to screen for quorum sensing inhibitory activity with compounds that inhibited Lux-based bioluminescence, 3-oxo-C_12_-HSL, C_12_-TMA and C_12_-TOA were mixed with *Gaussia* luciferase in the supernatant of *Gluc*-expressing *S. aureus* to investigate whether they directly influenced the GLuc biochemical reaction. They were also tested on cultures of constitutively bioluminescent *S. aureus* expressing *lux* to compare the effect on *lux-*based bioluminescence.

As shown in Figure 6, neither 3-oxo-C_12_-HSL nor C_12_-TMA significantly affected GLuc bioluminescence compared with a 1% v/v DMSO solvent control. However, when these two compounds were incubated with Lux + *S. aureus* at 5 µM, light output was significantly reduced. This confirmed that *lux-*based bioluminescence was directly inhibited by these compounds. C_12_-TOA was not shown to significantly decrease either GLuc or *lux*-based bioluminescence at both concentrations tested.

### 3.6. Evaluation of USA300 agrP3 GLuc as a Screen for agr Inhibitors Related to 3-oxo-C12-HSL

To evaluate whether *S. aureus* USA300 *agrP3* GLuc could be used to screen analogues of 3-oxo-C_12_-HSL, this strain was cultured in the presence or absence of a range of concentrations of 3-oxo-C_12_-HSL, C_12_-TMA and C_12_-TOA and bioluminescence read after 6 h. As shown in Figure 7, addition of 5 µM or 10 µM of 3-oxo-C_12_-HSL significantly reduced bioluminescence compared to the 1% DMSO control. The addition of C_12_-TMA and C_12_-TOA also reduced bioluminescence which confirmed findings of *agr* inhibition by these compounds reported by Murray et al. (2014). This demonstrates that *S. aureus* USA300 *agrP3* GLuc can be used to screen inhibitors of *agr*-dependent quorum sensing, where screening by Lux is precluded due to direct inhibition of Lux bioluminescence.

## 4. Discussion

Antibiotic resistance in many bacterial pathogens including *S. aureus* is increasing and antibiotics of last resort are increasingly used to treat these infections, which in turn is leading to the emergence of resistance to these high-value antibiotics. There is an urgent need for novel therapeutics which do not succumb to conventional mechanisms of antibiotic resistance. One method is through attenuation of bacterial virulence. In *S. aureus*, pathogenicity is largely controlled by the *agr* quorum sensing system, therefore this is a potential target for antivirulence drugs [13,41]. It is therefore important to develop simple methods for monitoring of *agr* activity in order to facilitate screening for inhibitors and gain information about *agr* activation in infection models in vivo. In this study we have demonstrated that *Gaussia* luciferase can be used as a reporter for *agr* dependent quorum sensing in *S. aureus*. The ability of GLuc to report *agr* activation in vitro was demonstrated by comparing bioluminescence over time with OD_600_. This showed that bioluminescence increased in the late log phase of growth when *agr* is activated [11]. This confirmed that the reporter functioned as anticipated and is likely to be useful for monitoring temporal *agr* activation both in vitro and potentially in vivo in experimental animal infection models.

To demonstrate that GLuc activation is under the control of the *agr* system, the response to synthetic AIP-1 was first evaluated. Activation of the *agr* system in vitro in wild type *S. aureus* strains by addition of exogenous synthetic AIP-1 can be difficult to observe because of other regulators of the *agr* system. For example, CodY represses *agr* activation during exponential growth in rich media, preventing *agr* activation before depletion of key nutrients [42]. Therefore, in order to evaluate the response of the reporter to exogenous AIP-1, *agrA* and *agrC* which make up the two-component signal transduction system were introduced into USA300 Δ*agr* P3-GLuc on plasmid pSKermP2. This plasmid encodes *agrC* and *agrA* under the control of the P2 promoter of the *agr* system [36]. Complementation of *agrA* and *agr*C enabled the Δ*agr* strain to respond to exogenous AIP-1. This was shown by the increased light output on provision of AIP-1 compared with the DMSO control.

It was demonstrated that for USA300 *agr*P3-GLuc the expression of *Gaussia* luciferase was dependent on activation of the *agr* QS system. This was further validated by addition of a competitive antagonist of the *agr* system. Addition of (Ala^5^)AIP-1 reduced bioluminescence. This confirmed that inhibition of the *agr* system also inhibited GLuc activation, which was consistent with the control of light output through activation of QS. Similar results were obtained with C_12_-TMA and C_12_-TOA which are noncompetitive, negative allosteric modulators of AgrC [40].

This work also demonstrates the potential of the GLuc reporter in identifying inhibitors of other *agr* system components. Ambuic acid, an AgrB inhibitor, was tested against USA300 *agr*P3-GLuc. Ambuic acid has been tested previously in in vivo models with an *agr*P3-*lux* reporter where real time reduction of *agr* activity was seen [15]. This shows that GLuc can be used as a reporter to screen for *agr* inhibitors and evaluate their potential as prophylactic and therapeutic agents.

Previously, bacterial *lux-*based reporter systems have been used to study *agr* quorum sensing however it has been demonstrated that some *agr* inhibitors can interfere with the luciferase activity giving false results. Here it was demonstrated that 3-oxo-C_12_ HSL interfered with the *lux* signal however did not interfere with GLuc bioluminescence. It was shown that QS inhibitors which have an effect on *lux*-based bioluminescence could be reliably assayed using GLuc, demonstrating that it was a useful alternative output for bacterial reporter systems. Additionally, the output from *lux-*based reporters may be affected by metabolic effects decreasing the availability of bioluminescence substrates such as FMNH_2_ at the point of assay, therefore any reduction in bioluminescence could be due to metabolic factors rather than a reduction of *agr* activity [43]. GLuc does not require cellular cofactors during assay therefore directly provides information about *agr* activation, avoiding the reduction in specificity associated with the loss of metabolites. This work demonstrates that *agr*P3 GLuc reporters can be used as a useful alternative or adjunct to screens for inhibitors of the *agr* system in vitro.

Together these data show that the *agr*P3-GLuc reporter can be activated by exogenous cognate AIP-1 and inhibited by antagonists of the *agr* system similar to *agr*P3-*lux* and *agr*P3-*blaZ* reporters [36,39,44]. This highlights the use of the *Gaussia* luciferase as a novel reporter of *agr* activity in *S. aureus* in vitro and for potential screening for activators and inhibitors of the *agr* system.

## Figures and Tables

**Figure 1 sensors-20-04305-f001:**
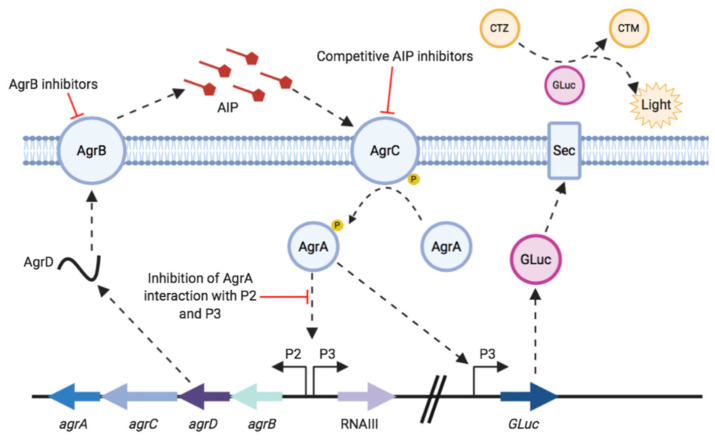
Outline of the staphylococcal *agr* system and the P3 GLuc reporter. Critical control points of the *agr* system, which are amenable to targeting by inhibitors, are indicated by blunt arrows. CTZ represents coelenterazine and CTM coelenteramide. Adapted from [13].

**Figure 2 sensors-20-04305-f002:**
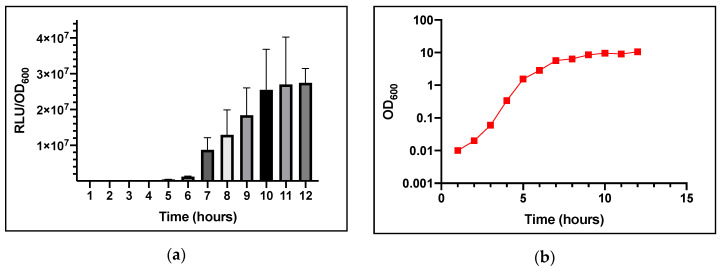
Bioluminescence from *S. aureus* strain USA300 *agr*P3-GLuc over time. (**a**) Bioluminescence output. (**b**) Bacterial growth (OD_600_). Data are representative of three technical repeats and three biological repeats for each sample with error bars displaying standard deviations.

**Figure 3 sensors-20-04305-f003:**
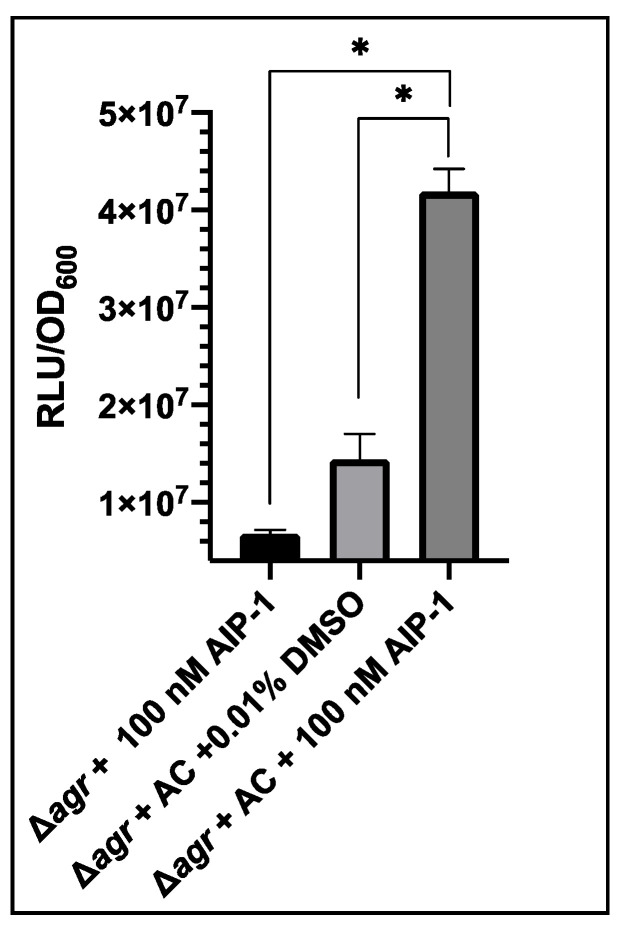
Impact of the addition of exogenous AIP-1 on bioluminescence produced by *S. aureus* USA300 Δ*agr* P3-Gluc pSKermP2 *agr*AC. Δ*agr* represents USA300 Δ*agr* P3-GLuc and Δ*agr* + AC represents USA300 Δ*agr* P3-Gluc pSKermP2 *agr*AC. The data represent three technical repeats and three biological repeats for each sample and error bars displaying standard deviation. * *p* < 0.01.

**Figure 4 sensors-20-04305-f004:**
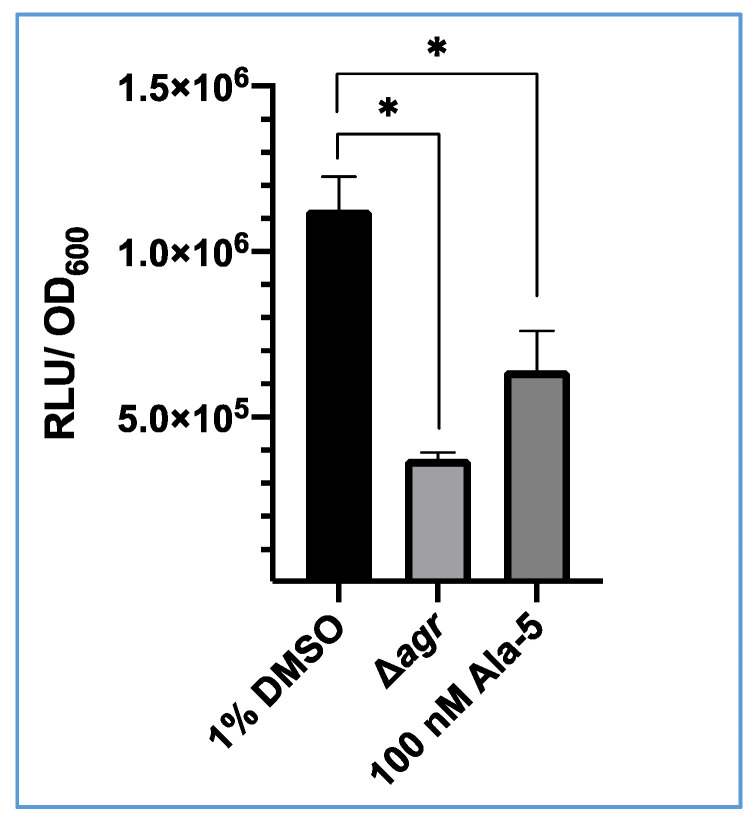
Impact of Ala^5^(AIP-1) on bioluminescence output from *S. aureus* USA300 *agr*P3-GLuc. The 1% v/v DMSO represents USA300 *agr*P3 GLuc with 1% DMSO, Δ*agr* represents USA300 Δ*agr* P3-GLuc and 100 nM Ala-5 represents USA300 *agr*P3 GLuc with addition of 100 nM (Ala^5^)AIP-1. The data shown are the means with error bars displaying standard deviation of triplicate technical repeats and is representative of biological triplicate repeats. * *p* < 0.01.

**Figure 5 sensors-20-04305-f005:**
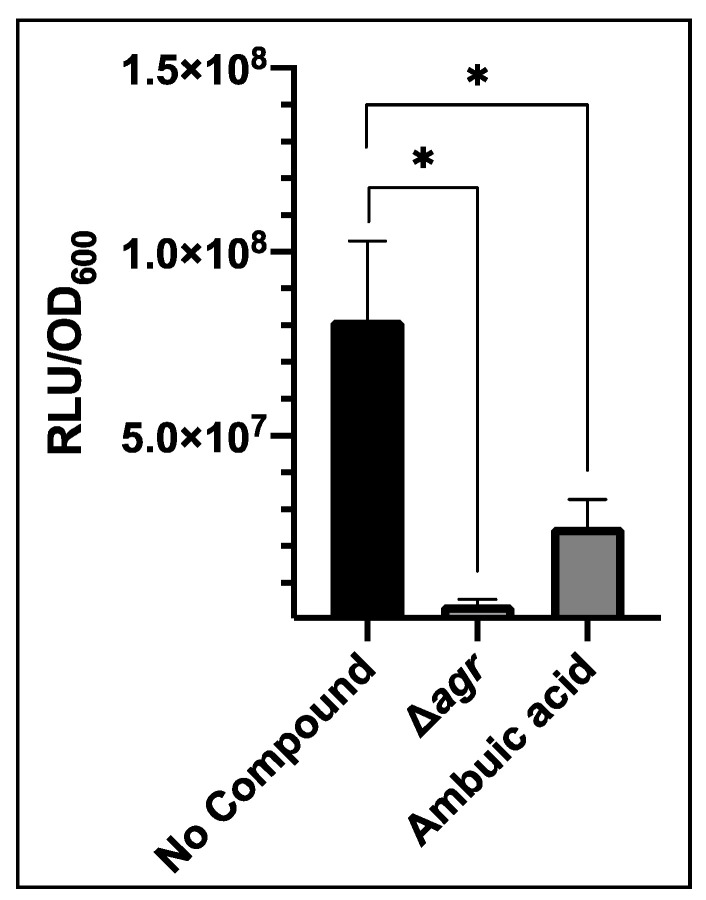
Impact of the AIP biosynthesis inhibitor ambuic acid (40 µM) on bioluminescence output from *S. aureus* USA300 *agr*P3-GLuc. ‘No compound’ represents USA300 *agr*P3 GLuc without inhibitor, Δ*agr* represents USA300 Δ*agr* P3-GLuc without inhibitor and ambuic acid represents USA300 *agr*P3 GLuc with the addition of 40 µM ambuic acid. The graph displays the mean of three technical and three biological repeats with error bars displaying standard deviation. * *p* < 0.01.

**Figure 6 sensors-20-04305-f006:**
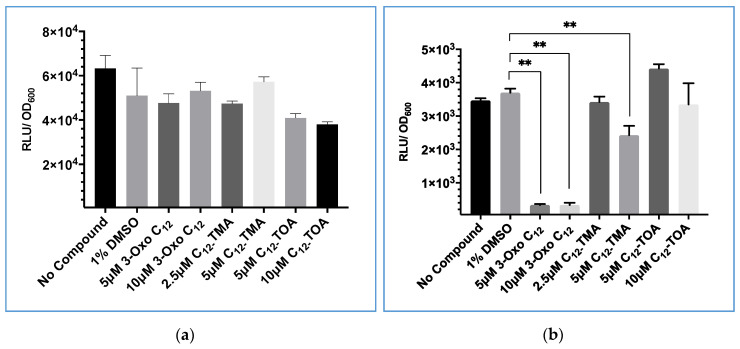
Comparison of the effect of 3-oxo-C_12_-HSL, C_12_-TMA and C_12_-TOA on *S. aureus* constitutively expressing bioluminescence based on (**a**) GLuc or (**b**) Lux. The data shown are the means with error bars displaying standard deviation of triplicate technical repeats and are representative of biological triplicate repeats. ** *p* < 0.05.

**Figure 7 sensors-20-04305-f007:**
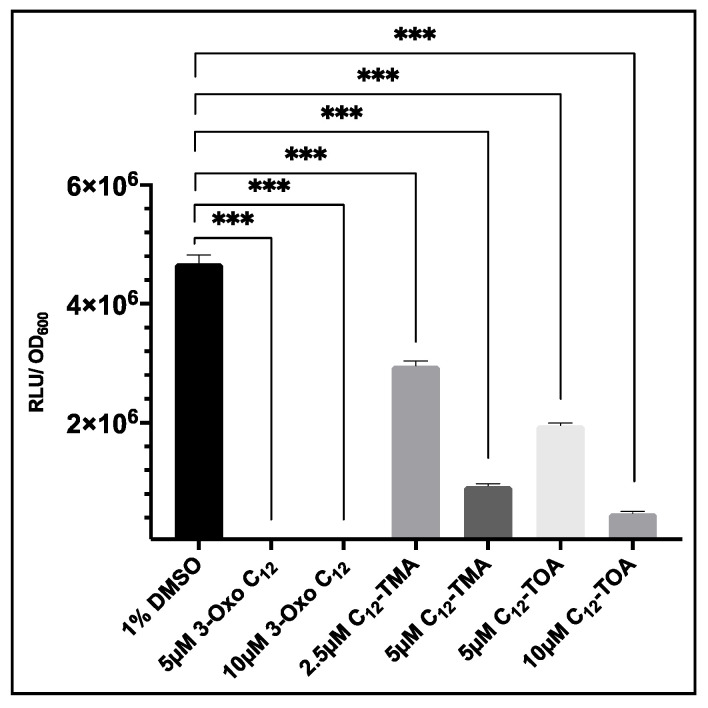
The effect of 3-oxo-C_12_-HSL, C_12_-TMA and C_12_-TOA on *agrP3-*GLuc expression in USA300 *agrP3* GLuc. For 3-oxo-C_12-_HSL, low luminescence values of 1.9 × 10^4^ RLU/OD600 (5 µM) and 2.45 × 10^4^ RLU/OD_600_ (10 µM) were recorded. The data show the means with error bars displaying standard deviation of triplicate technical repeats and are representative of biological triplicate repeats. *** *p* < 0.0001.

**Table 1 sensors-20-04305-t001:** Bacterial strains used in this study.

Strain	Description	Reference
*S. aureus* 8325-4	Derivative of *S. aureus* NCTC 8325	[32]
*S. aureus* RN4220 *att*B2	Cloning intermediate	[33]
*S. aureus* USA300 JE2	Plasmid-cured derivative of the CA-MRSA strain USA300 LAC	[34,35]
*S. aureus* RN4220 *pXyl/tet: SpA leader: GLuc*	Constitutively expressing SpA leader:GLuc	This study
*S. aureus* RN4220 *Pxyl/tet:::luxABCDE*	Constitutively bioluminescent strain	[20]
*S. aureus* USA300 *agr*P3 Gluc	USA300 with chromosomal integration of SpA leader and GLuc under the control of *agr*P3	This study
*S. aureus* USA300 Δ*agr* P3 Gluc	USA300 Δ*agr* with chromosomal integration of SpA leader and GLuc under the control of *agr*P3.	This study
*S. aureus* ROJ143	ROJ143 carrying plasmid pSKermP2 *agr*C1 *agr*A	[36]
*S. aureus* USA300 Δ*agr* P3 Gluc pSKermP2 *agr*C1 *agr*A	USA300 Δ*agr* P3 Gluc carrying pSKermP2 agrC1 *agr*A	This study
*E. coli* DC10B	Cloning intermediate *dc*	[37]

**Table 2 sensors-20-04305-t002:** Primers used in this study. All primers were sourced from Sigma Aldrich (UK).

Primer	Sequence	Function	Reference
CT12 (F)	CTAGTAGGAGGAAAAACATATGATGACTTTACA	Amplification of SpA:GLuc	This study
CT13 (R)	ATTTGTCGACCTCAGGAGAGCGTTCACC		
CT14 (F)	AGTGAATTCCCGGGGATCCGACACGTCGACCCTCACTG	Amplification of *agr*P3	This study
CT15 (R)	CTCCTACTAGCCATCACATCTCTGTGATCTAG		
CT16 (F)	GATGTGATGGCTAGTAGGAGGAAAAACATATGATG	Amplification of SpA:GLuc	This study
CT17 (R)	TCGATAAGCTTGGCTGCAGGATTTGTCGACCTCAGGAG		
OU9R10	CATACTACATATCAACGAAATCAG	Forward primer at RN4220 *att*B2 integration site	[33]
SCV4	ACCCAGTTTGTAATTCCAGGAG	Reverse primer at 5′ end of pLL102 to RN4220 *att*B2	[33]
SCV8	GCACATAATTGCTCACAGCCA	Forward primer at 3′ end of pLL102 to RN4220 *att*B2	[33]
OU9R7	ATGGGTGGTAAAACACAAATTTC	Reverse primer at RN4220 *att*B2 integration site	[33]

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
