# Peer review of "Gaussia Luciferase as a Reporter for Quorum Sensing in Staphylococcus aureus"

_sensors, 2020, doi:10.3390/s20154305_

Round 1
Reviewer 1 Report
The letter, entitled "Gaussia luciferase as a reporter for quorum sensing in Staphylococcus aureus", described Gaussia luciferase has a biosensor for the agr system in S. aureus. Authors widely demonstrated the efficiency of agrP3-Gluc fusion using specific inhibitors, mutants and complementation.
Major comments
- I agree that “GLuc enabled quantitative assessment of agr activity” in vitro.
However, I am not entirely convinced that this system is more suitable than others.
Especially, the agr system was already studied using another reporter gene, lux. Indeed, a similar agrP3-lux fusion was used to identify quorum sensing inhibitors and to monitor agr induction in murine abscesses. Here argP3-Gluc was only tested in TSB medium. Authors should directly compare both reporter gene fusions to support their statements.
Authors also argue that GLuc “poses no metabolic burden on the cell” (Lines 79-80) as it does not required ATP. Nevertheless, GLuc is a secreted protein, translocated by the Sec system, which is ATP-dependent. Moreover, they did not check the impact of the strong synthesis (P3 promoter) of GLuc on the secretion of other proteins using the Sec system.
Lines 277-279. Authors stipulated that some agr inhibitors can interfere with the lux system giving false results, but no details are given. Please add references. Could these inhibitors also affect GLuc activity?
- Line 175 "which is representative of agr expression in real time". It could be interesting to monitor the expression of RNAIII by Northern blot in similar conditions (Figure 2), notably to monitor the delay between expression and detection.
- Could authors indicate if Gluc is suitable for translational fusions? Knowing that the fusion protein has to be secreted.
Minor comments
- Figure 1. Please cite the reference(s) used to draw this illustration.
- Introduction. The function of RNAIII, which is under the control of P3 promoter, is not described.
- Line 118. Plasmidpa
- The agr mutant is not indicated in Table 1 or described in Materials and Methods.
- agrP3 or P3, please homogenize
- Figure 5. Could authors comment on the slight induction of P3 promoter observed in the agr mutant in presence of AC +0.01% DMSO?
- Lines 244-46. An efficient reporter system already exists, agrP3-lux
Author Response
We do not claim that our Gluc-based reporter system is more suitable than other reporters for all applications. In fact, in our cover latter we explicitly say that it “complements current Lux-based reporters of bacterial gene expression.” It however provides a facile tool which can be used where Lux-based reporters have proven inadequate.
We fully agree with Reviewer 1 that agrP3-lux reporters are very useful for a number of applications. Indeed, own agrP3-lux reporters were contemporaneous with the first ones described and have been extensively used by our group for over 15 years. Our deep experience with these reporters have shown us that Lux is a fantastically useful technology. However, its reliance on unimpeded metabolism at the point of assay, to provide reduced flavin and tetradecanal substrates, makes it prone to give false positives when screening for inhibitors of agr. It is precisely this aspect of Lux technology that prompted the development of an alternative, complementary reporter readout that is independent of the energy status of the cell, which we report in the current manuscript. We have reworded the MS to emphasise that Gluc is an alternative/adjunct to Lux for some applications.
We agree with the reviewer that expression of any heterologous protein as a reporter can impose some bioburden on the host cells. The MS have been modified so as not to claim Gluc expression is without burden.
We have added extra data to the MS to show the direct inhibition of Lux signal by small molecules (Figure 6) and have added citations to studies that previously reported inhibitors of the S. aureus agr system that showed direct Lux inhibition (Qazi et al 2006 and Murray et al. J. Med. Chem. 2014, 57, 2813−2819). These data are accompanied by directly comparable Gluc data which show that Gluc is not inhibited by these molecules. In these examples, Gluc and not Lux give a more reliable readout of agr inhibition.
We do not have Northern blot data alongside Gluc readouts, however, we know that the agrP3-Gluc readout shows similar induction kinetics to agrP3-Lux, nevertheless we have taken out references to ‘real time’ readout.
We know that we can successfully create translational fusions to GLuc but this is outside of the current MS and we believe would distract from the main point of the paper.
All minor comments are addressed in the revised MS, including a more detailed description of agr in the introduction to include RNAIII function and citing references used to support Figure 1; consistent use of agrP3, commenting on background signal where the transcriptional activator AgrA is present and including all strains in Table 1.
Reviewer 2 Report
Major comments:
the AgrC antagonist used in this study is not described in sufficient detail nor specified how it is produced. This is an important detail and must be included to ensure reproducibility of this study.
line 184: 'to be reported elsewhere' > this must be avoided and the antagonist must be well described and characterized
Minor comments:
line 172: does > was
Author Response
Reviewer 2 is not happy that we show data using a small-molecule competitive inhibitor of agr that has not, as yet, been fully disclosed. In response to these comments we have removed all reference to this inhibitor and instead replaced with equivalent data derived using another competitive inhibitor (Ala5(AIP-1)) which we have previously published (McDowell et al. 2001. Mol. Microbiol. 41, 503–512).
One of the authors of our original manuscript (Fabio Rui) was responsible for the synthesis of the undisclosed inhibitor in our original MS and no other part of this work, so has been removed as an author.
We trust this fully answers this reviewer’s major comment and we have amended the single minor comment in the revised manuscript.
Reviewer 3 Report
The activity of Gaussia luciferase as a reporter for quorum sensing in Staphylococcus aureus was studied in this manuscript. Several modified strains were obtained to test the functionality of the model.
The following comments are made:
- Line 46. Briefly explain Figure 1, so that the reader understands how the agr system works.
- Line 88 to 92. Put this phrase in Material and Methods. Which of all the McDougal USA300 clinical strains did you use?, because you did not use an ATCC strain?
- Line 118. What does plamidpa mean?
- Line 174. Say the meaning of RLU
- Line 177. Figure 2 and all the figures, the scale of the Y axis is not adequate since it starts at 0 and the values ​​are from ~ 106 to 107, so the values ​​do not match. Modify this in all Figures.
- Line 183. “both strains”, what are they?
- Line 188. What is the proportion of inhibition? Move what you said on line 195.
- Line 198. Figure 3, They are not WT strains, they are USA300 agrP3-Gluc and SH1000 agrP3-Gluc strains.
- Line 211. Figure 4, Same as the previous point, they are not WT strains.
10. Line 234. Figure 5, put the correct name of the strains or explain it in the figure footer. Since Δagr is USA300 Δagr P3-Gluc and Δagr + AC is USA300 Δagr P3-Gluc pSKermP2 agrAC. You should put the control of USA300 agrP3-Glu to compare the effect. Correct “100 nm AIP-1” must be
Author Response
The activity of Gaussia luciferase as a reporter for quorum sensing in Staphylococcus aureus was studied in this manuscript. Several modified strains were obtained to test the functionality of the model. The following minor comments are made:
1. Line 46. Briefly explain Figure 1, so that the reader understands how the agr system works.
We have expanded the description and citation of the agr system in the introduction (Page 2 lines 48-55) to include the role of RNAIII as the effector of the QS response and signposted clearly to Figure 1 which shows a graphical representation of the agr system.
2. Line 88 to 92. Put this phrase in Material and Methods. Which of all the McDougal USA300 clinical strains did you use?, because you did not use an ATCC strain?
The phrase has been moved to the Materials and Methods. We have cited the full name and references to the USA300 clone we used in Table 1 Row 3.
3. Line 118. What does plamidpa mean?
This was a typographical error, which has been corrected
4. Line 174. Say the meaning of RLU
RLU is relative light units and defined in methods page 4 line 161
5. Line 177. Figure 2 and all the figures, the scale of the Y axis is not adequate since it starts at 0 and the values are from ~ 106 to 107, so the values do not match. Modify this in all Figures.
The large dynamic range of the data that contains very low values for uninduced strains means that, the very lower values are difficult to see on our plots. We have tried various presentation methods and believe that we have plotted the data in the clearest possible way. Where lower values are not easily read, we have explicitly stated these values in the text (e.g. for Fig. 4) or legend (e.g. Fig. 7).
6. Line 183. “both strains”, what are they?
7. Line 188. What is the proportion of inhibition? Move what you said on line 195.
8. Line 198. Figure 3, They are not WT strains, they are USA300 agrP3-Gluc and SH1000 agrP3-Gluc strains.
These comments refer to section of the MS that has been removed as it contained the data for the unpublished inhibitor that reviewer 2 was not happy with.
9. Line 211. Figure 4, Same as the previous point, they are not WT strains.
This figure has become figure 5 in the revised MS; Reference to WT has been removed and figure legend revised to fully identify all strains
10. Line 234. Figure 5, put the correct name of the strains or explain it in the figure footer. Since Δagr is USA300 Δagr P3-Gluc and Δagr + AC is USA300 Δagr P3-Gluc pSKermP2 agrAC. You should put the control of USA300 agrP3-Glu to compare the effect. Correct “100 nm AIP-1”
This figure has become Figure 3 in the revised MS We have corrected the AIP-1 error and fully revised the figure legend to made explicit which strains are being referred to. The purpose of this figure is to indicate that our reporter responds directly and specifically to exogenous AIP1. Hence the use of an agr mutant which cannot synthesise or sense AIP-1, and a complemented mutant which can sense AIP1 but not synthesise it. Figure 2 in the revised MS already shows activation of the P3 reporter by the unmutated USA300 agrP3-Glu by its own AIP-1.

Round 2
Reviewer 1 Report
I only have minor comments:
- In Figures 2, 3 and 5 authors indicate that “data represent three technical repeats and three biological repeats for each sample and error bars displaying standard deviation”, but in Figures 4, 6 and 7 “the data show are the means with error bars displaying standard deviation of triplicate technical repeats and is representative of three biological triplicate repeats.” Could authors please explain this discrepancy especially since it is more relevant to compare biological repeats than technical repeats.
- Line 68. generation, ,
- Line 81. “some compounds” Please give one or two examples?
- Line 92-93. “without the need for energy from live cells during assay. [23].” Please rephrase this sentence as an ATP-dependent export system is still required.
- Lines 228-231, 247 and 275. agr in italic
- Figure 4. ΔAgr
- Figure 6 and Lines 330-331. Authors should discuss the observed effects of C12-TOA on the GLuc-based reporter.
- Line 365. both USA300
Reviewer 3 Report
- In the new version Figure 3 remains the error of "100nm", it must be 100nM.
- In Figure 7, again you put 0 on the Y axis, you should remove it or put 0.9X106 or 9x105, since that represents the distance you put between one value and another.
3. There was no response to observation 11 of the previous comments: To make your system more robust, it is recommended to compare it with the expression of some S. aureus adhesin or toxin to see if the effect is comparable with the bacteria genes. This would enhance your work.
